# Reviewer 1：

1. Can you provide a detailed explanation of the differences between SimCC and traditional heatmaps?

Re 1: Dear professor, firstly, thank you very much for your valuable advices. In the traditional 2D heatmap method, the position of each keypoint is represented as a two-dimensional Gaussian distribution heatmap, where a high-probability peak is generated at a certain location on the heatmap, indicating that this position is most likely to be the keypoint. The SimCC method treats keypoint prediction as two independent classification tasks, one for the horizontal coordinate and the other for the vertical coordinate. SimCC divides each pixel into multiple smaller bins, achieving sub-pixel level localization accuracy. The traditional 2D heatmap method suffers from significant quantization errors when the input resolution is low, requiring additional upsampling layers and post-processing steps to reduce these errors. In contrast, the SimCC method achieves sub-pixel accuracy even at lower resolutions by dividing each pixel into multiple bins, thereby reducing quantization errors. To improve localization accuracy, the traditional 2D heatmap method typically requires multiple complex upsampling layers, such as deconvolution layers, as well as additional post-processing steps. The SimCC method accomplishes coordinate classification tasks with only two lightweight classification heads (each containing only a linear layer), eliminating the need for complex upsampling layers and post-processing steps, thereby simplifying the model architecture and improving efficiency.

2. How is formula (9) obtained through cosine similarity? Please provide a detailed process.

Re 2： Cosine similarity is used to measure the directional similarity between two vectors. Its formula is:

$$\cos\theta = \frac{\mathbf{A} \cdot \mathbf{B}}{|\mathbf{A}| \cdot |\mathbf{B}|}$$

$\mathbf{A}$ and $\mathbf{B}$ are two vectors. $\mathbf{A} \cdot \mathbf{B}$ represents the dot product (or inner product) of vectors $\mathbf{A}$ and $\mathbf{B}$. $|\mathbf{A}|$ and $|\mathbf{B}|$ are the magnitudes of vectors $\mathbf{A}$ and $\mathbf{B}$, respectively. The cosine similarity value ranges from 0 to 1, indicating the degree of similarity between the two vectors, with values closer to 1 implying greater directional similarity.

In Equation 8, $A_t$ and $B_t$ represent two feature vectors at time $t$. The dot product of the feature vectors is calculated element-wise, and the total dot product of the two vectors is

obtained by summing up the products of corresponding elements. Simultaneously, the magnitude of each vector is calculated by taking the square root of the sum of the squares of its components. The detailed calculation process is as follows:

Dot Product Calculation: Multiply each corresponding component of the two feature vectors and sum them up to get the numerator, $\sum_{t=1}^{n} A_t \times B_t$.

Magnitude Calculation: Calculate the magnitude of vectors **A** and **B** by taking the square root of the sum of the squares of their components, $\sqrt{\sum_{t=1}^{n}(A_t)^2}$ and $\sqrt{\sum_{t=1}^{n}(B_t)^2}$, respectively.

Normalization: Finally, the dot product (numerator) is normalized by dividing it by the product of the two vector magnitudes using Equation 9 to obtain the final cosine similarity value.

3. How to introduce a Transformer-based high-resolution structure in HRFormer?

Re 3：HRFormer draws on the multi-resolution parallel design concept of High-Resolution Network (HRNet). Throughout the entire model, HRFormer maintains a high-resolution feature stream and gradually adds medium- and low-resolution feature streams, which are connected in parallel. The feature representations of each stream are updated at their respective resolutions using multiple Transformer blocks. Meanwhile, a convolutional multi-scale fusion module enables information exchange between different resolutions, thereby achieving multi-scale feature modeling.

HRFormer introduces a local window self-attention mechanism at each resolution. Specifically, the representation map is divided into a set of non-overlapping small image windows, and self-attention operations are performed independently within each window.

# Reviewer 2：

1. How does the LGAG self-attention mechanism improve the accuracy of key point detection?

Re 1: LGAG employs 3×3 group convolutions instead of conventional 1×1 convolutions for processing the gating signal and input feature maps. This choice allows the model to capture a larger local context around each key point, making the attention mechanism more sensitive to spatial details. As a result, the model can focus on more relevant spatial regions, reducing the likelihood of missing key features.

By combining features from skip connections and upsampled features using learned attention coefficients, LGAG progressively refines the feature maps. This process enhances the representation of critical spatial features while suppressing irrelevant information, leading to more precise localization of key points.

2.  What are the advantages of soft label coding over hard label coding?

Re 2: Soft labels can express the similarity and ordinal relationships between classes, helping the model understand the connections between adjacent classes. This is particularly useful for tasks with natural order, such as age estimation. Soft labels lead to a smoother variation in the loss function, aiding faster convergence during training, enhancing generalization ability, and reducing the risk of overfitting. By distributing probability across multiple classes, soft labels reduce the impact of single mislabeled instances, making the model more resilient when dealing with noisy or uncertain labels. The probability distributions generated by soft labels better reflect data uncertainty, thereby improving the reliability of the model's prediction confidence.

# Reviewer 3:

1.  The disadvantages of the proposed method and the direction of the next work should be explained in the conclusion.

Re 1: Our revised conclusions are as follows:

Although the proposed improved RTMPose model demonstrates significant performance improvements in dance pose evaluation, it still has certain limitations. The model's accuracy declines when handling complex dance movements and dense occlusion scenarios, indicating that further optimization is needed for high-difficulty pose estimation tasks. Second, while the model's real-time performance has improved, the demand for computational resources remains high when processing multi-person, multi-action scenarios. Optimizing the model structure to further reduce computational complexity and improve the efficiency of multi-person pose estimation.

2.  This study may have different designs in comparison to some previous studies, but how are these differences significant? The author needs to highlight and clarify them in innovation point.

Re 2: The main innovations of this study compared to previous works are highlighted in the following aspects:

Improved Keypoint Representation: Our study introduces the Large-kernel Grouped Attention Gate (LGAG) to enhance the representation of keypoints. Unlike traditional attention mechanisms, LGAG captures more complex spatial dependencies and nonlinear features, particularly useful for complex dance movements.

Transformation of Pose Estimation Task: We reframe the 2D pose estimation problem as a coordinate classification task using the SimCC method, instead of relying on heatmap-based methods. This approach reduces quantization errors and improves localization accuracy.

Model Efficiency and Scalability: By replacing the standard 7x7 convolutional layers with three 3x3 convolutional layers, we significantly reduce the model's parameters and computational cost while maintaining high accuracy. This structural optimization enhances the model's scalability and efficiency.

These innovations collectively contribute to a more efficient and accurate system for evaluating dance standardization scores, addressing limitations found in previous methods such as high computational complexity and lower accuracy in dynamic pose scenarios."

3. There is no need to repeatedly define abbreviated forms of terms, such as 'Large-kernel Grouped Attention Gate (LGAG)', which appears several times in the text, which is unnecessary.

Re 3: Thank you for pointing that out. I have reviewed and revised the document to remove the redundant definitions of abbreviated terms such as 'Large-kernel Grouped Attention Gate (LGAG)'. Now, the terms are defined only once and used consistently throughout the text. I appreciate your feedback and attention to detail.

# Research on an Improved RTMPose Model for Evaluating Dance Standardization Scores

Di Cao
*Civil Aviation College*
*Shenyang Aerospace University*
Shenyang, China
714121874@qq.com

Qixuan Sun
*College of Artificial Intelligence*
*Shenyang Aerospace University*
Shenyang, China
sunqixuan@stu.sau.edu.cn

*Tong Cui
*College of Artificial Intelligence*
*Shenyang Aerospace University*
Shenyang, China
*ct61ct61@126.com

*Abstract*—**Human pose estimation is a critical technology in computer vision that enables machines to understand and interpret human movement. One important application of this technology is the standardized analysis of dance movements. In this paper, we improve the RTMPose model by optimizing keypoint representation through convolutional layers and Large Kernel Gated Attention Units (LGAG), transforming 2D pose estimation into a coordinate classification task. The LGAG self-attention mechanism enhances keypoint detection accuracy. Additionally, the loss function based on SimCC, combined with soft label encoding, further optimizes the model's performance. Comparative experiments conducted on the COCO, MPII, and our collected standard dance datasets demonstrate that our model significantly improves human pose estimation performance and effectively achieves standardized dance movement scoring. This highlights its potential for various applications in sports, health, education, and human-computer interaction.**

*Keywords—Dance Standardization Evaluation, Pose Estimation, LGAG, RTMPose, SimCC*

## I. INTRODUCTION

In recent years, human pose estimation has become a popular topic in computer vision, offering significant benefits and potential applications for improving human life. One important application of human pose estimation is the standardized analysis of dance movements. However, this field faces unique challenges due to the complexity of dance movements and the uncertainty of human biological characteristics. Accurate keypoint recognition of human dance movements can enable standardized detection, which could have a substantial impact in various fields, including sports, health, dance education, and human-computer interaction.

Human pose estimation can be categorized into four main approaches: methods based on pose graphs (further divided into regression-based and heatmap-based methods); detection-based methods (which include top-down and bottom-up approaches); graph convolution-based methods; and Transformer-based algorithms. Toshev [1] introduced Deep Pose, a method for human pose estimation based on deep neural networks (DNNs), achieving high-precision pose estimation through cascaded DNN regressors. Wang [2] developed the Hourglass network specifically for human pose estimation, incorporating a stacked concept that effectively captures multi-level features by integrating multi-scale information. Chen [3] proposed the

Cascade Pyramid Network (CPN), a novel architecture consisting of two stages: Global Net, which handles simple keypoints, and Refine Net, which focuses on occluded or invisible keypoints. This method significantly improved performance on the COCO dataset. Li [4] introduced the Residual Log-Likelihood Estimator (RLE), which learns distribution shifts to simplify the training process. The RLE-based approach increased mean Average Precision (mAP) by 12.4 on the MSCOCO dataset. Wei [5] proposed the Convolutional Pose Machines (CPM), which directly operates on confidence maps from the previous stage, improving keypoint localization accuracy by progressively refining predicted heatmaps. Cao [6] presented Open Pose, which utilizes Part Affinity Fields (PAFs) to associate body parts with individuals in images through a non-parametric representation. The method demonstrated significant improvements in performance and accuracy by refining PAF processing. Ke [7] et al. proposed the High-Resolution Network (HRNet) for human pose estimation, which maintains high-resolution representations throughout the process, avoiding the common issue of recovering high resolution from low-resolution inputs. Sun [8] introduced a method that combines regression with heatmap generation by integrating heatmap operations to directly compute keypoint coordinates, balancing the strengths of both approaches. Li [9] introduced PG-GCN, which uses human keypoint information to guide the learning of graph convolutional networks, enhancing the model's recognition capability by capturing structured information among keypoints. Wu [10] improved ST-GCN by integrating an enhanced Detection Transformer (DETR) structure, effectively addressing issues related to Non-Maximum Suppression and anchor generation, thereby improving node spatial relationship utilization and detection accuracy in human pose estimation. Moon [11] introduced MSA R-CNN, a model that combines human detection and keypoint localization, utilizing multi-scale information to achieve better performance in 2D multi-person pose estimation. Wang [12] proposed a human pose detection technique that combines top-down and bottom-up approaches, extracting keypoint and object location features through Mask R-CNN and integrating Part Affinity Fields (PAFs) and confidence maps from CNNs for precise pose recognition. Ulya [13] introduced the HiroPoseEstimation model, which evaluates datasets using a bottom-up and top-down approach with Keypoint Mask R-CNN and a single-stage encoder-decoder

model, showing good performance on the proposed dataset. Kreiss [14] proposed PifPaf, a novel bottom-up multi-person 2D human pose estimation method that uses Part Intensity Fields (PIF) to locate body parts and Part Affinity Fields (PAF) to connect them into a complete human pose. Zhang [15] introduced an improved 2D human pose estimation method that incorporates attention mechanisms and hard sample mining techniques to address low accuracy and long inference times in multi-person scenarios. Chen [16] proposed a method that learns human keypoints through convolutional networks, obtaining keypoint attention regions for classification. Li [17] introduced a Pose-Oriented Transformer (POT) combined with an Uncertainty-Guided Refinement Network (UGRN), where POT uses pose-oriented self-attention mechanisms and distance-based position embeddings, and UGRN further refines hard-to-predict joints. Li [18] proposed VTTransPose, which incorporates dual attention modules in the TransPose model and replaces basic blocks in the third subnet with V modules to enhance joint feature representation and network recognition performance. Ma [19] developed the Token-Pruned Pose Transformer (PPT) model, which significantly reduces computational costs by focusing self-attention calculations on selected tokens after identifying the rough body area. Ren [20] introduced the Distilling Pruned-Token Transformer (DPPT), leveraging the output of the pre-trained TokenPose model to supervise PPT's learning process, addressing performance degradation caused by background token pruning. Jiang introduced RTMPose, a model that achieves accurate detection and localization of human keypoints through an efficient network architecture that significantly reduces computational load and model size while maintaining accuracy, thanks to improved heatmap regression and keypoint offset regression techniques. Jiang [21] proposed RTMPose, by leveraging improved heatmap regression and keypoint offset regression techniques, RTMPose achieves precise detection and localization of human keypoints.

In human pose estimation methods, the pose graph regression approach directly regresses the positions of keypoints and can utilize simple network structures. However, it has weaker capabilities in handling complex poses and occlusions, and it struggles to capture global context, leading to lower accuracy. Heatmap-based methods can capture fine keypoint positions and improve accuracy and robustness through multi-scale feature fusion. These methods have high computational complexity, typically requiring substantial computational resources, which results in slower inference speeds. Additionally, there are memory consumption issues in generating and processing high-resolution heatmaps. Top-down detection methods achieve high accuracy by performing pose estimation within each detected bounding box individually. However, in multi-person scenarios, each person must be detected and estimated separately, leading to high computational costs and slower processing speeds. Bottom-up detection methods, on the other hand, detect all keypoints across the entire image and then assemble them, making them suitable for real-time applications. However, they struggle to correctly identify and assemble keypoints in densely populated or heavily occluded scenes. Graph Convolutional Network (GCN)-based algorithms can capture the complex dependencies between human keypoints and perform well in processing high-dimensional data. However, the construction and operation of graphs involve high computational complexity, making GCN models potentially slow in both training and inference. Transformer-based algorithms excel in capturing global context information and improve pose estimation accuracy through self-attention mechanisms. They perform exceptionally well in handling long-range dependencies and complex poses. However, the computational complexity is high, and Transformer architectures require significant computational resources and data for training, which can lead to overfitting or slower inference speeds.

To address these issues, we propose an improved attention mechanism based on RTMPose. By enhancing the convolutional layers following the Backbone and utilizing a Large-kernel Grouped Attention Gate (LGAG), we improve the model's ability to capture nonlinear features in human pose recognition tasks with higher efficiency. This method alleviates the high computational complexity and overfitting problems common to transformer architectures. In addition, we combine this approach with a feature vector-based action matching algorithm, resulting in a system capable of standardized dance scoring.

## II. RELATED WORKS

### A. Multi Pose Estimation Method

Bottom-up algorithms detect instance-agnostic keypoints within an image and then segment these keypoints to obtain human poses. The bottom-up paradigm is considered well-suited for crowded scenes because the computational cost remains stable regardless of the number of individuals. However, these algorithms typically require high input resolution to handle the varying scales of people, making it challenging to balance accuracy and inference speed.

On the other hand, top-down algorithms use off-the-shelf detectors to provide bounding boxes, then crop the human body to a uniform scale for pose estimation. The top-down paradigm has consistently dominated public benchmarks. The two-stage inference paradigm allows human detectors and pose estimators to use relatively low input resolution, which enables them to outperform bottom-up algorithms in terms of speed and accuracy in non-extreme scenarios (i.e., when the number of people in an image does not exceed six).

This study adopts a top-down approach primarily because it is better suited for handling the complex poses involved in dance movements. Dance actions are often precise and varied, requiring accurate localization, and tracking of different body parts. The top-down algorithm enhances keypoint detection accuracy by first using an off-the-shelf detector to accurately locate the human body, followed by pose estimation at a uniform scale. Additionally, since the number of participants in dance scenarios is typically small, the top-down approach can achieve high inference speed and accuracy with relatively low computational resource consumption. This is crucial for real-time dance pose analysis and standardized recognition. In this

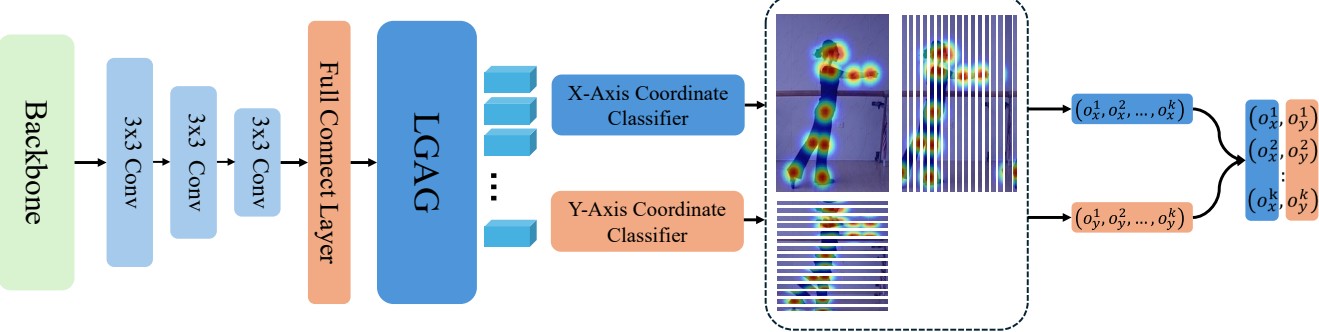

Fig. 1. The Proposed RTMPose Network Structure

study, we use RTMDet, a high-performance, low-latency single-stage object detector, as the human detector.

### B. Coordinate Classification

Traditional heatmap-based algorithms work by reducing the image resolution and then predicting heatmaps for human keypoints in a smaller resolution space. Each keypoint's heatmap represents a probability distribution across the image, with high-value regions indicating the likely locations of keypoints. However, this method struggles in scenes where keypoints are densely packed, especially in cases of complex human poses or severe occlusion, where the heatmap can become increasingly ambiguous.

In the traditional 2D heatmap method, the position of each keypoint is represented as a two-dimensional Gaussian distribution heatmap, where a high-probability peak is generated at a certain location on the heatmap, indicating that this position is most likely to be the keypoint. This method suffers from significant quantization errors when the input resolution is low, requiring additional upsampling layers and post-processing steps to reduce these errors. To improve localization accuracy, the traditional 2D heatmap method typically requires multiple complex upsampling layers, such as deconvolution layers, as well as additional post-processing steps.

Unlike traditional heatmaps, SimCC treats Human Pose Estimation (HPE) as two separate classification tasks: one for the x-coordinate and one for the y-coordinate. SimCC first deploys a CNN or Transformer-based backbone to extract potential keypoints. It then divides each pixel into multiple smaller bins, achieving sub-pixel level localization accuracy. By treating keypoint prediction as two independent classification tasks, SimCC can individually locate the x and y coordinates of each keypoint, thereby eliminating the need to maintain a two-dimensional feature map. This approach not only reduces quantization errors but also eliminates the need for complex upsampling layers and post-processing steps. Instead, SimCC accomplishes coordinate classification tasks with only two lightweight classification heads, each containing only a linear layer, thereby simplifying the model architecture and improving efficiency. As a result, SimCC achieves sub-pixel accuracy even at lower resolutions, making it more robust in handling complex scenarios with dense keypoints and occlusions.

### C. Transformer in Vision

The Transformer model was initially widely used in natural language processing (NLP) due to its exceptional self-attention mechanism and global feature capturing ability. It was later introduced into the computer vision field and gradually applied to pose estimation tasks. The self-attention mechanism of Transformers enables the establishment of long-range dependencies between different parts of an image, which is highly beneficial for modeling the relationships between human keypoints in pose estimation. Unlike CNNs, which rely solely on local receptive fields, Transformers can effectively capture spatial relationships between different parts of the human body.

HRFormer introduced a Transformer-based high-resolution structure specifically designed for dense prediction tasks, including pose estimation. HRFormer adopts the multi-resolution parallel design from HRNet, maintaining a high-resolution feature stream while progressively adding medium- and low-resolution streams. Each stream updates its features at its respective resolution using multiple Transformer blocks. A convolutional multi-scale fusion module facilitates information exchange between resolutions for multi-scale feature modeling. Additionally, HRFormer introduces local window self-attention, where the representation map is divided into non-overlapping windows, and self-attention is applied independently within each window. DETR (Detection Transformer) was initially proposed for object detection, but its global self-attention mechanism and direct regression of positions have also been applied to pose estimation. By directly regressing keypoint positions instead of relying on heatmaps, DETR has improved the accuracy of pose estimation to some extent. RTMPose combines the self-attention mechanism with a compact SimCC-based representation to capture keypoint dependencies. This significantly reduces computational load and allows for real-time inference with higher accuracy and efficiency. RTMPose employs a variant of the Transformer, the Gated Attention Unit (GAU), which offers faster speed, lower memory cost, and better performance compared to standard Transformers.

In this study, we optimized the 7x7 convolutional layers and GAU used in RTMPose. By improving the LGAG, we enhance the accuracy of RTMPose, particularly in cases where there is high similarity between different keypoints.

### III. METHOD

The improved RTMPose network structure includes two convolutional layers, a fully connected layer, and a Large-kernel Gated Attention Unit (LGAG) to optimize the representation of $K$ keypoints. The 2D pose estimation is then framed as two separate classification tasks for the x-axis and y-axis coordinates,

predicting the horizontal and vertical positions of the keypoints. The network architecture is illustrated in Fig. 1.

## A. SimCC

We adopt the coordinate classification approach referenced from SimCC, framing keypoint localization as a classification problem. The core idea is to divide the horizontal and vertical axes into equally spaced, numbered bins and discretize continuous coordinates into corresponding bin labels. The model is then trained to predict the bin in which the keypoint resides. By using a large number of bins, quantization error is reduced to a sub-pixel level.

For the backbone, we use CSPNeXt from RTMDet. In RTMPose, we replace the final layer of CSPNeXt with a 7x7 convolutional layer to obtain keypoint representations. However, instead of the 7x7 convolutional layer, we use three 3x3 convolutional layers, which reduces the number of parameters and computational cost by approximately 44.9%. Additionally, we incorporate the Exponential Linear Unit (ELU) activation function in the convolutional layers to enhance non-linearity, allowing the model to learn more complex feature representations. This stacked approach captures local features while gradually accumulating higher-level feature representations layer by layer. The three 3x3 convolutional layers are also easier to integrate with other network modules and optimize, offering strong adaptability and flexible application across different model architectures.

$$\text{ELU}(x) = \begin{cases} x & x > 0 \\ \alpha(\exp(x) - 1) & x \leq 0 \end{cases} \quad (1)$$

The feature map is processed through three 3x3 convolutional layers, resulting in a feature map of shape $(K, H, W)$, where $K$ is equal to the number of keypoints. The last two dimensions of the feature map are then merged, flattening the two-dimensional features into a one-dimensional representation, yielding a matrix of shape $(K, H \times W)$. A fully connected layer is then used to expand this one-dimensional keypoint representation to the desired dimensionality, which is controlled by a hyperparameter.

## B. Transformer

The performance of heatmap-based models improves as the feature resolution increases. To further enhance model performance, we employ a self-attention mechanism. Specifically, we utilize a variant of the Transformer, the LGAG. Compared to the standard Transformer and other variants like the Gated Attention Unit (GAU), LGAG offers faster speed, lower memory cost, and better performance.

LGAG incrementally integrates the feature maps with attention coefficients learned by the network, thereby enhancing the activation of relevant features and suppressing irrelevant ones. This method leverages gating signals derived from higher-level features to control the information flow at various stages of the network, which improves the accuracy of keypoint detection. The network structure of LGAG is shown in Fig. 2.

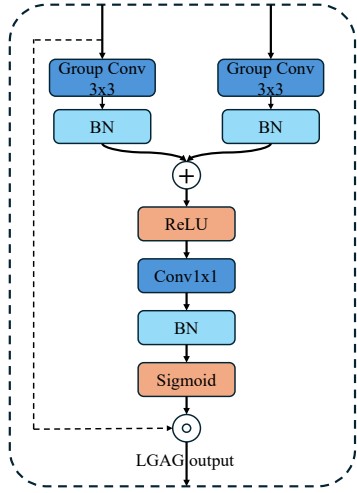

Fig. 2. Network structure of LGAG

In LGAG, the features g and x are processed separately by applying individual 3x3 Group Convolutions. These convolutional features are then normalized using Batch Normalization (BN) and combined through element-wise addition. The resulting feature map is activated by a ReLU layer. Following this, a 1x1 convolutional layer and another BN layer are applied to obtain a single-channel feature map. The single-channel feature map is then passed through a Sigmoid activation function to generate the attention coefficients. The output of this transformation is used to scale the input feature x through element-wise multiplication, producing the attention-gated feature $LGAG(g, x)$.

$$q(g, x) = R\left(BN\left[\left(GC_g(g)\right) + BN\left(GC_x(x)\right)\right]\right) \quad (2)$$

$$LGAG(g, x) = x \odot \sigma\left(BN\left[\left(q(g, x)\right)\right]\right) \quad (3)$$

where $BN$ stands for batch normalization, $GC_g$ and $GC_x$ represent 3x3 group convolutions, $R$ is the ReLU activation function, $\sigma$ is the Sigmoid activation function, and $\odot$ denotes element-wise multiplication.

## C. Loss Function

This study predicts pose keypoints based on the SimCC algorithm, which frames keypoint localization as a classification task for horizontal and vertical coordinates. To address the intra-class and inter-class relationships in keypoint localization during Pose Estimation, we adopt the soft label encoding method proposed in SORD [22]. This approach does not require any explicit modifications to the network architecture and can seamlessly incorporate metric penalties into the ground truth label representation to constrain these relationships between classes. Soft labels capture the similarity and ordinal relationships between classes, aiding the model in understanding connections between adjacent classes, which is especially beneficial for tasks like age estimation. They result in smoother loss variation, promoting faster convergence, better generalization, and reducing overfitting. By distributing probability across multiple classes, soft labels mitigate the impact of mislabeled instances, making the model more robust

to noisy or uncertain labels. Additionally, the probability distributions from soft labels better represent data uncertainty, improving the reliability of prediction confidence.

In this work, we use an unnormalized Gaussian distribution as the metric for inter-class distance.

$$\phi(r_t, r_i) = exp\left(\frac{-(r_t-r_i)^2}{2\sigma^2}\right) \tag{4}$$

$$y_i = \frac{exp(\phi(r_t,r_i))}{\sum_{k=1}^{K} exp(\phi(r_t,r_k))} \tag{5}$$

where $\phi(r_t, r_i)$ is a metric loss function of our choice that penalizes how far the true metric value of $r_t$ from the $r_i \in Y$.

In SimCC, the same encoding method is applied to the bins along both the horizontal and vertical directions. Therefore, for the standard deviation σ of the unnormalized Gaussian distribution.

$$\sigma = \sqrt{\frac{W_S}{16}} \tag{6}$$

where $W_s$ is the bin number in the horizontal and vertical directions, respectively. Therefore, the complete loss function for the model is defined as follows

$$Loss = -\sum_{n=1}^{N} \sum_{k=1}^{K} W_{n,k} \cdot \sum_{i=1}^{L} \frac{1}{W_s} \cdot V_i log(y_i) \tag{7}$$

where $N$ is the number of the person samples in a batch, $K$ is the number of keypoints. $W_{n,k}$ is a target weight mask to distinguish invisible keypoints. $V_i$ is the label value.

### D. Standardized detection of movements

To achieve this, we adopt a dance pose analysis method based on feature vector matching [23]. By comparing the action feature vectors with predefined standard feature vectors, this method evaluates the similarity between the current action and the standard action, thereby enabling standardized detection of movements.

Based on the human keypoint data obtained in Section II, every three feature points can define a feature plane, allowing the similarity between the dancer and the standard action to be attributed to the geometric similarity between objects. According to ergonomic principles, the human skeleton can be divided into seven feature planes, as illustrated in the Fig. 3. The analysis of movement standardization can thus be simplified into a comparison of the similarity between edge vectors within the same plane and the similarity of normal vectors between planes.

The calculation method is as follows: Using the human spine as the main axis, the spine serves as the z-axis of the spatial Cartesian coordinate system, with the x-axis and y-axis of the horizontal plane as the ground plane. Cosine similarity is used as the similarity function. By measuring the cosine of the angle between the inner products of two vectors in space, we can assess their similarity.

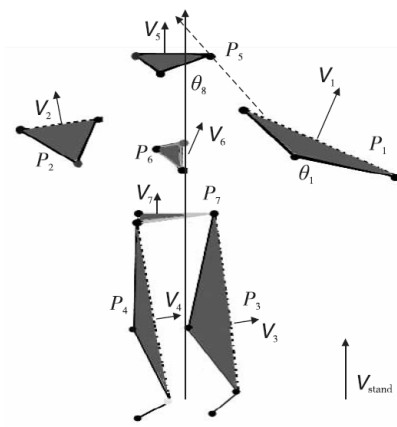

Fig. 3.   Feature Plane based on Keypoints in the Human Skeleton

The cosine similarity value ranges from 0 to 1, indicating the degree of similarity between the two vectors, with values closer to 1 implying greater directional similarity. Compared to Euclidean distance, cosine similarity focuses primarily on the angle between the two vectors, measuring the similarity in their direction in space without considering differences in body size, such as variations in arm length among different dancers.

$$Sim(\theta_i) = \frac{\sum_{t=1}^{n} A_t \times B_t}{\sqrt{\sum_{t=1}^{n}(A_t)^2} \times \sqrt{\sum_{t=1}^{n}(B_t)^2}} \tag{8}$$

where $Sim(\theta_i)$ represents the cosine similarity of the joint angles, with $\theta_i$ being the joint angle; $A_t$ and $B_t$ represent two feature vectors at time $t$. If the value is close to 1, it indicates that the dancer's actions are highly consistent with the standard actions, implying standardized movements. If the value is close to 0, it indicates a significant difference between the dancer's actions and the standard actions.

To address the differences in height, weight, arm length, and other physical attributes among different dancers, cosine similarity can be used to measure whether the range of motion of the limbs meets the standard. By measuring the similarity and difference between angles, the results can be obtained. The calculation formula is as follows:

$$Corr(U, V) = 1 - \left(\frac{arccos(Sim(\theta_i))}{\pi}\right) \tag{9}$$

where $U$ and $V$ are the two vectors being compared, and $Sim(\theta_i)$ is the cosine similarity of the angle $\theta_i$ between vectors $U$ and $V$. To calculate the difference in feature poses, for each synchronized time frame of the two video frames, the correlation coefficient of each action part is calculated. Then, the error of the correlation coefficient is calculated in the form of relative error. The calculation formula is as follows:

$$\Delta corr_t = \frac{corr_i - corr_j}{corr_i} \times 100\% \tag{10}$$

where $t$ is the time point of the dance action; $corr_i$ is the correlation coefficient of the standard dance action; $corr_i$ is the correlation coefficient of the dance action being evaluated. The

condition for error convergence is $\Delta \text{corr}_t \leq C$, where $C$ is the selected error threshold, which can be adjusted according to the actual dance standard requirements.

## IV. EXPERIMENT

We conducted experiments on a computer equipped with an Intel Core i7-12700 CPU, 32GB RAM, and an NVIDIA GeForce RTX 4090 GPU. The software environment is set up using the Ubuntu 20.04 LTS operating system and PyTorch 1.13. We also utilized MMPose (an open-source toolbox for pose estimation) and MMDet (a toolbox for object detection) to build the environment. For the specific experimental parameters, the learning rate was set to 0.000017, and the batch size set to 128. The model was trained for a total of 300 epochs.

### A. Dataset description

We used the collected standard dance movement video dataset as the training and testing dataset to evaluate our proposed method. Below is a description of the datasets used.

The COCO dataset contains extensive 2D human keypoint annotations and is well-suited for pose estimation tasks. It provides annotations for 17 keypoints, including the head, shoulders, elbows, knees, and other body parts, covering approximately 150,000 human instances across 250,000 images. COCO's human keypoint annotations are widely used for training and evaluating pose estimation models, supporting multi-pose and multi-view analysis.

The MPII dataset focuses on human pose estimation and provides over 40,000 human keypoint annotations across various everyday activities. MPII annotates 16 keypoints on the human body, covering a diverse range of activities and is one of the standard datasets for human pose estimation. Each image also includes activity labels and 2D skeleton models, supporting detailed pose analysis.

The collected standard dance movement video dataset was recorded using a 1080P 30FPS webcam, capturing a series of videos with each video lasting 2 minutes. A total of 300 videos were recorded, involving 30 students. The videos were extracted into images using OpenCV and annotated following the COCO format, including keypoints and bounding boxes for 17 body parts such as the head, shoulders, elbows, knees, and more.

### B. Ablation experiments

This section evaluates the effectiveness of our proposed model by clearly analyzing the contribution of each component to the overall performance, particularly the improved attention mechanism and the Backbone structure.

We compared the performance using the following four models:

Baseline Model: The original RTMPose model without modifications.

Improved Model 1: Introduces the LGAG without altering the Backbone structure.

Improved Model 2: Modifies only the final convolutional layer of the Backbone (changing from 7x7 to three 3x3 layers) without introducing LGAG.

Fusion Model: Incorporates both LGAG and the Backbone improvements.

These four models were compared using the fusion strategy on the COCO, MPII, and the collected standard dance movement video datasets. The results are presented in TABLE I. .

TABLE I.    ABLATION COMPARISON OF IMPROVED MODELS AND BASELINE MODEL

| Dataset | Method | | | |
|---------|----------|-------|----------|--------|
| | BaseLine | LGAG | Backbone | Fusion |
| COCO | 71.4 | 73.45 | 73.67 | 74.68 |
| MPII | 92.7 | 93.82 | 92.93 | 94.33 |
| Dance Dataset | 81.2 | 82.45 | 80.72 | 83.37 |

The experimental results demonstrate that both the LGAG and the Backbone improvements independently contribute to the model's performance. Specifically, introducing LGAG led to a 1.47 AP (Average Precision) improvement, while the Backbone modification alone resulted in a 0.67 AP increase. When both enhancements were combined, the overall AP improvement reached 2.36. This indicates that LGAG has a significant advantage in capturing nonlinear features, particularly when handling complex movements, and the improved model exhibits stronger expressiveness and robustness.

### C. Comparative experiments

We conducted extensive comparative experiments to validate the effectiveness of our model on the COCO, MPII, and the collected standard dance movement video datasets. The results are shown in TABLE II. .

Our method achieved the highest accuracy on the MPII and dance datasets, particularly excelling in the dance dataset, which involves complex movements. These results indicate that the proposed improvements significantly enhance the model's performance, especially in scenarios requiring precise and robust pose estimation. The superior performance on the dance dataset underscores the model's effectiveness in handling complex and dynamic poses, which are critical for applications in dance movement analysis.

TABLE II.    COMPARATIVE EXPERIMENT RESULTS

| Method | Dataset | | |
|--------|---------|------|---------------|
| | COCO | MPII | Dance Dataset |
| RTMPose | 71.4 | 92.7 | 81.2 |
| OpenPose | 56.3 | 88.8 | 74.36 |
| HRNet-W32 | 75.8 | 92.3 | 79.81 |
| **Our method** | 74.68 | 94.33 | 83.37 |

### D. Standardized scores of dance

In the field of dance, due to the fixed duration of standard dance movements, we can standardize the length of the dance video to be evaluated with the length of the standard video. In the experiment, we extract human keypoints from both the test video and the standard video to construct sequences and

calculate the feature vectors for each frame using the Method III approach. The accuracy score S for different methods in the context of dance standardization is calculated using the following formula:

$$S = 1 - \frac{1}{N}\sum_{i=1}^{N}|S_i - T_i| \qquad (11)$$

where N is the number of frames in the video. $S\_i$ and $T\_i$ are the feature vectors of the standard video frame and the test video frame, respectively.

This formula calculates the accuracy score S by measuring the difference between the feature vectors of the standard and test video frames, then averaging these differences across all frames. The score S reflects the degree of similarity between the evaluated dance movements and the standard movements, with higher scores indicating better alignment with the standard. The experimental results are shown in Fig. 4.

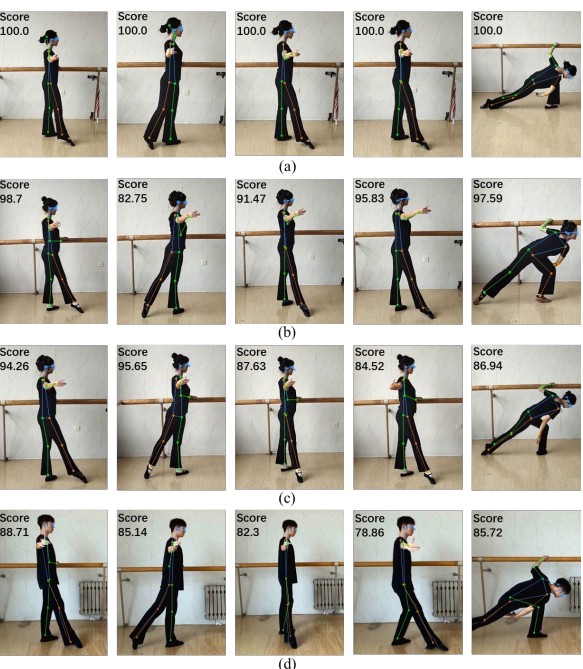

Fig. 4. Standardized score evaluation of dance (a) is the real-time score of the standard action, and (b), (c) and (d) are the real-time scores of the test video

## V. CONCLUSION

This study proposes an enhanced RTMPose model that improves keypoint representation by integrating convolutional layers with a Large-kernel Gated Attention Unit (LGAG). The approach transforms the 2D pose estimation problem into a classification task for the x and y coordinates. By utilizing the SimCC method, we treat keypoint localization as a classification problem, dividing the coordinate axes into evenly spaced, numbered bins to effectively minimize quantization error. Structurally, we replaced the 7x7 convolutional layer with three 3x3 convolutional layers, reducing the number of parameters and computational costs, and introduced the ELU activation function to enhance nonlinear feature learning. Additionally, by incorporating the LGAG self-attention mechanism, the model more effectively activates relevant features, improving keypoint

detection accuracy. Finally, we designed a SimCC-based loss function, integrating a soft label encoding method to further optimize keypoint localization, thereby enhancing the model's accuracy and robustness. This model advances potential applications in several areas, including dance training and education, artistic creation and choreography, competitive sports, and human-computer interaction. The model's accuracy declines when handling complex dance movements and dense occlusion scenarios, indicating that further optimization is needed for high-difficulty pose estimation tasks. Second, while the model's real-time performance has improved, the demand for computational resources remains high when processing multi-person, multi-action scenarios. Optimizing the model structure to further reduce computational complexity and improve the efficiency of multi-person pose estimation.

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
