# OpenReview forum: "Research on an Improved RTMPose Model for Evaluating Dance Standardization Scores"
_IEEE.org/ICIST/2024/Conference — IEEE ICIST 2024 Conference Submission_

### Official Review · Reviewer_pNXc · 2024-08-21
**This paper improves the RTMPose model by optimizing keypoint representation through convolutional layers and Large Kernel Gated Attention Units, transforming 2D pose estimation into a coordinate classification task. The LGAG self-attention mechanism enhances keypoint detection accuracy. Additionally, the loss function based on SimCC, combined with soft label encoding, further optimizes the model's performance. The topic of this paper is interesting. Below is a list of comments that should be taken into account further when revising the paper.**

**Rating:** 7
**Confidence:** 3

**Review:**

1. The disadvantages of the proposed method and the direction of the next work should be explained in the conclusion.
2. This study may have different designs in comparison to some previous studies, but how are these differences significant? The author needs to highlight and clarify them in innovation point.
3. There is no need to repeatedly define abbreviated forms of terms, such as ‘Large-kernel Grouped Attention Gate (LGAG)’, which appears several times in the text, which is unnecessary.

---

### Official Review · Reviewer_zdt4 · 2024-08-22
**This article is quite fascinating and of high quality.**

**Rating:** 7
**Confidence:** 3

**Review:**

This paper, " Research on an Improved RTMPose Model for Evaluating Dance Standardization Scores," proposes a neurodynamic approach to robust portfolio selection. First, the RTMPose model is improved by optimizing the key point representation through convolutional layers and Large Kernel Gated Attention Units (LGAG). Finally, the performance of the model is further optimized by using the loss function based on SimCC and soft label coding. The article has clear logic and organization, but there are still some problems. My specific feedback is as follows :1) How does the LGAG self-attention mechanism improve the accuracy of key point detection? 2) What are the advantages of soft label coding over hard label coding?

---

### Official Review · Reviewer_uGq4 · 2024-08-23
**Research on an Improved RTMPose Model for Evaluating Dance Standardization Scores**

**Rating:** 7
**Confidence:** 2

**Review:**

This study proposes an enhanced RTMPose model that improves key point representation by integrating convolutional layers with a Large Kernel Gated Attention Unit. The Large Kernel Gated Attention Units self-attention mechanism enhances key point detection accuracy. There are some problems that should be replied. Comments for this submission are given as follows:
1. Can you provide a detailed explanation of the differences between SimCC and traditional heatmaps?
2. How is formula (9) obtained through cosine similarity? Please provide a detailed process.
3. How to introduce a Transformer-based high-resolution structure in HRFormer?

---

### Decision · Program_Chairs · 2024-09-06

Accept (Oral)